# Efficient Non-DHT-Based RC-Based Architecture for Fog Computing in Healthcare 4.0

Indranil Roy [1,*], Reshmi Mitra [1], Nick Rahimi [2] and Bidyut Gupta [3]

[1] Department of Computer Science, Southeast Missouri State University, Cape Girardeau, MO 63701, USA; rmitra@semo.edu

[2] Department of Computer Science, School of Computing Sciences and Engineering, University of Southern Mississippi, Hattiesburg, MS 39406, USA; nick.rahimi@usm.edu

[3] Department of Computer Science, School of Computing, Southern Illinois University, Carbondale, IL 62901, USA; bidyut@cs.siu.edu

[*] Correspondence: iroy@semo.edu

**Abstract:** Cloud-computing capabilities have revolutionized the remote processing of exploding volumes of healthcare data. However, cloud-based analytics capabilities are saddled with a lack of context-awareness and unnecessary access latency issues as data are processed and stored in remote servers. The emerging network infrastructure tier of fog computing can reduce expensive latency by bringing storage, processing, and networking closer to sensor nodes. Due to the growing variety of medical data and service types, there is a crucial need for efficient and secure architecture for sensor-based health-monitoring devices connected to fog nodes. In this paper, we present publish/subscribe and interest/resource-based non-DHT-based peer-to-peer (P2P) RC-based architecture for resource discovery. The publish/subscribe communication model provides a scalable way to handle large volumes of data and messages in real time, while allowing fine-grained access control to messages, thus enabling heightened security. Our two-level overlay network consists of (1) a transit ring containing group-heads representing a particular resource type, and (2) a completely connected group of peers. Our theoretical analysis shows that our search latency is independent of the number of peers. Additionally, the complexity of the intra-group data-lookup protocol is constant, and the complexity of the inter-group data lookup is $O(n)$, where n is the total number of resource types present in the network. Overall, it therefore allows the system to handle large data throughput in a flexible, cost-effective, and secure way for medical IoT systems.

**Keywords:** Healthcare 4.0; fog computing; IoT devices; peer-to-peer network; non-DHT-based; interest/resource-based P2P

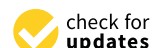



## 1. Introduction

In the past two decades, electronic gadgets have become an integral part of human life, with the incorporation of artificial intelligence and machine learning making them "smart" devices. One area where this is particularly relevant is in healthcare, where smart devices are being used for health monitoring, diagnosis, and even treatment. The interconnection of these medical devices, known as the Internet of Medical Things (IoMT), allows remote monitoring capabilities with fewer hospital visits for patients. The market for smart health devices is expected to grow at an average rate of 16.2% between 2020 and 2027. Advancements in the IoMT have made preliminary diagnostics possible at the patient's home, shifting healthcare from a hospital-centric to a home-centric service.

The introduction of Healthcare 4.0 standards in the healthcare industry has led to the integration of various technologies such as the Internet of Things (IoT), big data analysis, artificial intelligence (AI), robotics, continuous data sensing, cloud computing, and real-time actuators to create digital healthcare products, technologies, services, and enterprises. This shift in the healthcare industry requires various applications to meet the dynamic demands

of the industry, and involves procuring and developing well-equipped, efficient, concise, and cost-effective solutions to the problems of the healthcare industry. These technologies are leading towards revolutionizing healthcare systems, enabling them to provide real-time support to patients, aiming at the early prediction and prevention of diseases.

Despite the advantages, there are significant challenges associated with this paradigm shift. The architecture of healthcare applications and devices that rely on sensors collecting data on clouds can be constrained by delays in communication between sensors and clouds, and the high cost of storing and transferring large amounts of data on cloud-based storage. Additionally, the data associated with these healthcare applications may be time-sensitive, along with privacy concerns, especially for storage with third-party cloud service providers. Furthermore, network failures or congestion can disrupt communication between sensors and cloud services, which could risk a patient's life. The security of personal health information has previously been addressed in several studies [1–4].

To solve these latency, privacy, and scalability issues, fog computing combined with cloud-based data analytics for healthcare data has become a reasonable solution. Fog computing acts as an interface between IoT sensors/actuators and cloud services, and uses edge devices/networking devices with limited resources that may limit their ability to analyze large datasets and add to processing time. It leverages a distributed data-processing architecture closer to the sensor nodes, significantly reducing latency, where complex jobs are divided into simpler ones to reduce processing time and efficiently analyze the data received. However, fog computing performs initial processing on sensed data by bringing a complete or partial set of cloud services to the edge/fog devices.

Due to resource constraints, not every fog device can render complete services. Therefore, a single cloud-based service or API may be located at a single edge device or distributed among different devices. This placement of services ensures prompt data analysis/processing at nearby nodes. This approach helps to lower the latency, enhance security, and reduce the overall cost of data processing and storage. In summary, fog computing is a way to bring data processing and analysis closer to the source, using edge devices and networking devices with limited resources. This allows for faster data analysis and a reduction in the amount of data stored in the cloud, and can help overcome the challenges of resource discovery and management in the healthcare industry. However, there are still challenges that need to be addressed, such as ensuring the availability of services and self-reorganization infrastructure. These can be solved using *event-based publish/subscribe* systems and P2P overlays based on Distributed Hash Tables (DHTs) [5].

In this context, we state now what is meant by a DHT-based system. A distributed hash table (DHT) is a class of decentralized distributed system that provides a lookup service similar to a hash table; (key, value) pairs are stored in a DHT and any node in the system can find the value associated with a given key efficiently. Responsibility for maintaining the mapping from keys to values is distributed among all nodes in the system in such a way that a change in the set of participants does not cause any major disruption. This is why DHT can scale to very large numbers of nodes and can efficiently handle node arrivals, departures, and failures. Any system designed based on the idea that DHT is known as a structured system. There exists another type of decentralized distributed structured system that does not use the idea of DHT; one such system is known as a Residue Class (RC)-based system. RC-based architecture is covered in detail in the next section.

To fulfill the demands for scalability and flexibility, event-based publish/subscribe systems contain a variety of features. Three main parts make up a publish/subscribe system: *a publisher, a subscriber, and an event-notification service or broker*. A registered event triggers the publisher to notify the broker, which is the workhorse for the entire architecture. Subscribers can sign up with the broker by indicating their interest in a certain category of events. To the interested subscribers, the broker *asynchronously distributes* the events produced by the publisher. Decoupling between other parts of these systems is introduced via the use of an intermediary broker. In terms of synchronization, time, and location, the publishers and subscribers are unrelated [6]. The publish/subscribe architecture allows

for decoupling between publishers and subscribers, making the system more flexible. Publishers do not need to interact with each subscriber individually, and subscribers do not need to check for new events periodically. This type of architecture is well suited for large-scale, many-to-many interactions, and allows for the easy integration of new components. Additionally, the system is also flexible, as a publisher of one or more events can also act as a subscriber for other events, and vice versa.

The literature suggests a variety of publish/subscribe implementation designs. These are peer-to-peer, distributed, and centralized. A centralized broker is suggested by centralized solutions for event distribution. However, when the number of events rises, it experiences inherent scaling problems. The scalability issues of a centralized method are solved by distributed infrastructure. This is appropriate for the quick and effective distribution of temporary data. Peer-to-peer infrastructure, the third option, offers greater flexibility, scalability, and adaptability. Peers are utilized in this instance to store subscriptions and route events to the correct subscribers.

Due to its flexibility and scalability, peer-to-peer (P2P) overlay networks are preferable solutions for large-scale applications. Mobile networks can use P2P system principles. P2P systems treat each node equally and directly promote resource sharing between these nodes. As a result, the system is more resilient because the loss of one node will not affect the others. P2P networks also can accommodate users' dynamic nature and are economical since resources are shared. They may be divided into two types of systems: structured and unstructured. Performance problems plague unstructured peer-to-peer systems such as Kaaza [7] and Gnutella [8]. Distributed hash tables (DHTs), which offer remarkable load balancing, search efficiency, minimal overhead, and fault tolerance under high network dynamics, are typically used in structured peer-to-peer networks. Therefore, it is thought that DHT is a preferable option for implementing a publish-subscribe system. Content Addressable Networks (CAN) [9], Chord [10], Pastry [11], and Tapestry [12] are a few examples of well-known DHT-based peer-to-peer systems.

Application-layer implementations are the only ones allowed for traditional publish/subscribe systems. These systems have several difficulties, including a lack of self-organization, the absence of effective matching algorithms, and scale problems. Implementing a publish/subscribe system over a P2P overlay network can address these innate problems. The literature has some implementations of publish-subscribe systems based on DHT, including PastryStrings [13], Scribe [14], Meghdoot [15], and Hermes [16]. For publish-subscribe applications, each of them uses a separate DHT.

DHTs incur operational costs due to the churn issue, which demands a workaround for delivering a reliable data query service. Several significant research [13,15,17–19] have addressed creating hybrid systems to combine the benefits of both structured and unstructured designs with significant trade-offs. An interest/resource-based non-DHT-based structural design technique has also received a lot of attention [20,21]. Alongside attempting to lessen the complexity of churn handling, it offers the benefits of DHT-based systems. In the current study, we propose a publish/subscribe and interest/resource-based non-DHT fog-computing architecture that enables effective resource sharing for sensor data analysis. For the design of the architecture, we took into account the interest/resource-based non-DHT-based architecture proposed in [20,21].

*Our Contribution*

Our main objective is to show the superiority of our *publish/subscribe and interest/resource-based non-DHT fog-computing architecture* over publish/subscribe DHT-based architectures from the viewpoints of search latency and data-lookup complexity. We have considered a number theoretic approach to building the architecture. The following facts support our decision to consider such an architecture: (1) In contrast to any structured DHT-based network, our overlay network's search latency is independent of the total number of peers present, (2) the complexity of the intra-group data-lookup protocol is constant, while the complexity of the inter-group data lookup is $O(n)$, where n is the total number of resources types present in the network.

The organization of the paper is as follows. In Section 2, we talk about the difference between DHT-based and RC-based architecture, and some related preliminaries of the proposed RC-based architecture. In Section 3, we present the Publish/Subscribe and RC-Based P2P Architecture for Fog Computing, the data-lookup protocols. In Section 4 we describe the comparison of our proposed architecture and its performance with some noted DHT-based systems. Finally, Section 5 concludes.

## 2. Distributed Hash Table P2P vs. RC-Based P2P

In this section, we go over the functionality, benefits, and challenges of DHT-based and Residue Class (RC)-based P2P architecture. This section explains why the RC-based P2P model was selected for P2P fog computing as opposed to alternative DHT-based P2P models. We also present theoretical results in support of RC-based P2P architecture for the resource/interest paradigm.

### 2.1. DHT-Based P2P

The term "structured P2P systems" refers to all DHT-based P2P systems. In a structured peer-to-peer system, the data items are given keys, and a graph is created that connects each key to the node that holds the appropriate data. There are several P2P algorithms based on DHT, including Chord [10], Can [9], Pastry [11], Tapestry [12], and others. Distributed Hash Tables (DHTs) must first be understood before we can examine P2P that is DHT-based. A unique data structure known as a hash table may map keys to values. It utilizes a unique algorithm known as the hash function, which accepts the original key as input and produces a key that is the distinct numerical representation of the original key. The value that corresponds to the number key is mapped. As a result, the hash table stores data as (key, value) pairs among millions of peers using a distributed hash table (DHT). Every peer can use a key to query the database, which then returns the key's value. A query is resolved in a DHT-based peer-to-peer system via a limited number of message exchanges among peers. Just a select few of the other peers—not all of them—are known to each peer.

DHT also employs a few churn-management techniques. All the peers have organized themselves into a ring in a circular DHT (like a Chord). A pair of (key, value) values and a distinct ID from the ID space are given to each peer. Moreover, only the peer's immediate predecessor and successor are known to each peer (and have IP addresses). When a peer inside the circular DHT receives a query message asking for information about a value connected to a certain key, it first determines if it oversees the value of the key being requested. If it does, it unicasts the information to the peer that originated the inquiry message. If not, it would, depending on the key, pass the query message to either its successor or predecessor. In this manner, a query message is sent through peers in a circular DHT until a query hit is made. Since n is the total number of peers in the network, we can state that the time complexity for a search in DHT-based P2P is $O(\log n)$. Nevertheless, maintaining DHT and dealing with the churn issue is a difficult process that demands a lot of work.

### 2.2. RC-Based P2P Architecture [20]

**Definition 1.** *A resource is defined as a tuple of the characters $< R_i, V >$, where $R_i$ stands for a resource's type and V for its value. Please note that a resource might have several values.*

For illustration, let V symbolize a specific actor and $R_i$ denote the resource category "movies". Therefore, $< R_i, V >$ represents movies (some or all) acted by a particular actor V.

**Definition 2.** *In a peer-to-peer system with n different resource kinds, let S be the set of all peers (i.e., n distinct common resources). Then $S = \{C_i\}, 0 \le i \le n - 1$, where $C_i$ represents the subset of peers that have the same resource type $R_i$. This subset $C_i$ is referred to as group i in this study. Furthermore, we presume that $C_i^h$ is the first peer among the peers in each group $C_i$ to join the system. The group-head of group $C_i$ is referred to as $C_i^h$. The following $two-level$ overlay architecture as shown in Figure 1 has been proposed in [20].*

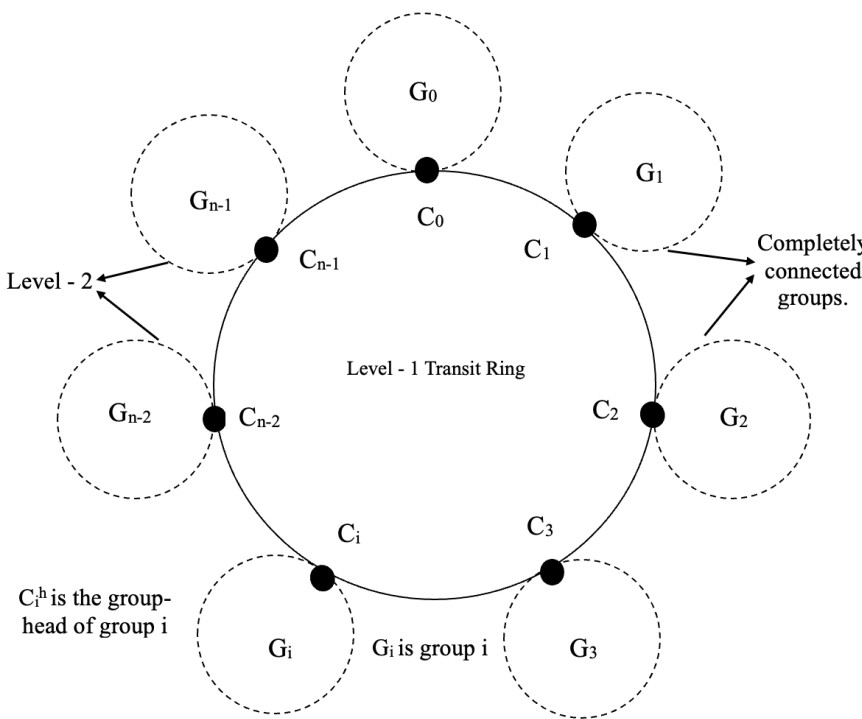

**Figure 1.** A two − level RC-based structured P2P architecture with n distinct resource types.

1. At level 1, there is a ring network consisting of the peers $C_i$ ($0 \leq i \leq n - 1$). The ring has n peers, which corresponds to the number of different resource kinds. This ring network is known as a transit ring since it is utilized for quick data search.

2. There are n totally linked networks (groups) of peers at level 2. Each of these groups, say $G_i$, is made up of the peers of the subset $C_{R_i}, 0 \leq i \leq n - 1$, in such a way that all of the peers ($\in C_{R_i}$) are logically connected and the network has a diameter of 1. Each $G_i$ has a group-head $C_i^h$ that connects it to the transit ring network.

3. Each peer on the transit ring network maintains a global resource table (GRT) that consists of n number of tuples. GRT contains one tuple per group and each tuple is of the form <Group-Head Logical address, IP address>, where Group-Head Logical Address refers to the architecture. Additionally, Resource Code is the same as the group-head logical address.

4. Each group-head $C_i^h$ also maintains a local resource table (LRT) that consists of k number of tuples, where k is equal to the number of members present in that group $G_i$. LRT contains a tuple of the form <Group member Logical address, IP address>. This LRT is also maintained by all the group-members of $G_i$.

5. Any communication between a peer $C_i \in$ group $G_x$ and $C_j \in$ group $G_y$ takes place only through the corresponding group-heads $C_x^h$ and $C_y^h$.

### 2.2.1. Relevant Properties of Modular Arithmetic

Consider the set $S_n$ of non-negative integers less than n, where $S_n$ = {0, 1, 2, . . . (n − 1)}. This is known as the residue set or residue classes (mod n). In other words, each integer in $S_n$ represents a residue class (RC). These residue classes are denoted by the symbols [0], [1], [2], . . . , [n − 1], where [r] = {a: a is an integer, a $\equiv$ r (mod n)}.

For example, for n = 3, the classes are:

[0] = {. . . , −6, −3, 0, 3, 6, . . . }

[1] = {. . . , −5, −2, 1, 4, 7, . . . }

[2] = {. . . , −4, −1, 2, 5, 8, . . . }

Thus, any class r (mod n) of $S_n$ can be written as follows:

[r] = {. . . , (r − 2n), (r − n), r, (r + n), (r + 2n), . . . , (r + (j − 1).n), (r + j.n), (r + (j + 1).n), . . . }

A few relevant properties of the residue class are stated below.

**Lemma 1.** *Any two numbers of any class r of $S_n$ are mutually congruent.*

**Proof.** Let us consider any two numbers N′ and N″ of class r. These numbers can be written as:

$$N' \equiv r (mod\, n); \text{ therefore,} (N' - r)/n = \text{an integer, say I}' \tag{1}$$

$$N'' \equiv r (mod\, n); \text{ therefore,} (N'' - r)/n = \text{an integer, say I}'' \tag{2}$$

Using (1) and (2) we obtain the following,

$$(N' - N'')/n = ((N' - r) - (N'' - r))/n = I' - I'' = \text{an integer.} \tag{3}$$

Therefore, N′ is congruent to N″; that is, N′ ≡ N″ (mod n);
Additionally, N″ ≡ N′ (mod n) because congruence relation (≡) is symmetric. Hence, the proof.  □

2.2.2. Assignments of Overlay Addresses

Assume that in a resource/interest-based P2P system, there are n distinct resource types. Please note that n can be set to an extremely large value a priory to accommodate many distinct resource types. Consider the set of all peers in the system given as S = {$C_{R_i}$} (($0 \leq i \leq n-1$)). Additionally, as mentioned earlier, for each subset $C_{R_i}$ (i.e., group $G_i$) peer $C_i$ is the first peer with resource type $R_i$ to join the system.

In the suggested overlay architecture [20–22], the positive integers belonging to distinct classes are utilized to determine the following parameters:

1.  Logical addresses of peers in a subset $C_{R_i}$ (i.e., group $G_i$): It will be shown how to use these addresses to support the claim that all peers ($\in G_i$) are (logically) directly linked to one another, producing an overlay network of diameter 1. Each $G_i$, as used in graph theory, is a whole graph.
2.  Identifying which peers on the transit ring network are neighbors with one another.
3.  Identifying each distinct resource type with a unique code.

The assignment of logical addresses to the peers at the two levels and the resources happen as follows:

1.  At level 1, the smallest non-negative number (r) of the residue class r (mod n) of the residue system, $S_n$ is assigned to each group-head $C_r^h$ of group $G_r$.
2.  At level 2, the group $G_r$ (i.e., the subset $C_{R_r}$) will be formed by all peers with the same resource type $R_r$, with the group-head $C_r^h$ connected to the transit ring network. Given to each new peer that joins group $G_r$ is the group membership address (r + j.n), where j is 1, 2, 3 . . . .
3.  Resource class $R_r$ possessed by peers in $G_r$ is assigned the code r which is also the logical address of the group-head $C_r^h$ of group $G_r$.
4.  A corresponding tuple of <Group-Head Logical Address, IP Address> is added to the global resource table (GRT) each time a new group-head joins.

**Remark 1.** *GRT remains sorted with respect to the logical addresses of the group-heads.*

**Definition 3.** *Two peers $C_i^h$ and $C_j^h$ on the ring network are logically linked together if (i + 1) mod n = j.*

**Remark 2.** *The last group-head $C_{n-1}^h$ and the first group-head $C_0^h$ are neighbors based on Definition 3. It justifies that the transit network is a ring.*

**Definition 4.** *Two peers of a group $G_r$ are logically linked together if their assigned logical addresses are mutually congruent.*

**Remark 3.** *The diameter of the transit ring network is $n/2$.*

**Lemma 2.** *Each group $G_r$ forms a complete graph.*

**Proof.** A pair of peers in a group $G_r$ are said to be logically connected by Definition 4, if their assigned logical addresses are compatible with one another. Additionally, from Lemma 1, we see that any two numbers of any class r of $S_n$ are mutually congruent. Hence, every peer has direct logical connectivity with every other peer in the same group $G_r$. Thus, the evidence. □

2.2.3. Salient Features of Overlay Architecture

We summarize the salient features of this architecture.

1. It is a two − level hierarchical overlay network architecture with a structured network at each level.
2. A group-head address would be identical to the resource type held by the group using modular arithmetic described in Section 2.2.1.
3. Unlike distributed hash table-based works that are now in use, some of which put a ring network at the center of their proposed design [10,23], the number of peers on the ring is equal to the number of different resource kinds.
4. Assume in general that there are already i group-heads ($C_0, C_1, \ldots C_{i-1}$) in the ring. The address i will then be given to the following peer joining the system as the group leader with resource type i. As an illustration, the sixth group-head joining the system will have the logical address 5 and the resource type code 5, respectively.
5. The diameter of the transit ring network is $n/2$. Please note that in any P2P network, the total number of peers N >> n.
6. In level 2, every overlay network is fully connected, i.e., in terms of graph theory, it is a full graph made up of the group peers. Its diameter is therefore just 1. The design provides the lowest feasible search latency inside a group due to its smallest diameter (in terms of overlay hops).

2.2.4. Fault Tolerance of the Architecture

To achieve fault tolerance, we assume that an existing group-head always saves a copy of the list of all peers in this group to the peer with the next logical address. In the architecture, we assume the following fault model. Any peer in any group may be faulty. First, let us consider that in a group, its group-head becomes faulty. After detection, it will be replaced by the peer with the next logical address in the group; the logical address of the faulty group-head now becomes the new logical address of this peer and this peer will start acting as the new group-head. Therefore, in GRT, only the IP address of this new group-head will be entered replacing the IP address of the faulty group-head. The new group-head will unicast the modified GRT to every other group-head. It will also save a copy of the updated list of peers to the peer with the next logical address. Next, let us consider that a peer in a group other than the group-head is faulty. All that is required is to delete the entry of the faulty peer in the list of peers and this will be done by the group-head; the group-head will save a copy of this updated list to the peer with the next logical address. It may be noted that in the present research, peer movements in and out of the architecture (i.e., churn) are not considered because the application may require stable peers. Therefore, the process of the replacement of a faulty peer may be viewed as if a peer has left the system (somewhat similar to churn). Please note that the structure of the group remains a connected network.

### 2.2.5. Scalability of the Architecture

It should be observed that in every overlay network, there are very few different resource kinds n, in comparison to the number of peers. A very large value of n can be chosen during the design process to handle a very large number of potential resource types, preventing the risk of having to modify the architecture when new groups are formed (if needed in the future). If new groups are established in the future with new resource types in the system, additional residue classes in sequence will be accessible for their addressing. This implies that if at first there are few resource types present, just the first few residue classes will be used initially for addressing. As an illustration, suppose that n is originally set to 1000. There are 1000 potential residue classes, starting with [0], [1], [2], [4], [5], . . . , [999]. The residue classes [0], [1], and [2] will be used to address the peers in the three unique groups if there are originally only three groups of peers and three different resource types present. The leftover classes [3] and [4] will be used to address the peers in the two new groups in order of their entering the system if later two new groups with two new resource kinds are generated. As we can see, any group size is unlimited since any residue class may be used to conceptually address an infinite number of peers that have a shared interest. As a result, there are no drawbacks to the suggested architecture's scalability.

### 2.3. Comparison of Distributed Hash Table P2P vs. RC-Based P2P

As the average number of hops needed for each search in Chord [10] is N/2, where N is the total number of peers in the system, following the chord is not done since it is exceedingly inefficient in big peer-to-peer systems. According to [20], the average number of hops needed for each search (on the ring network) is n/2, where n is the total number of different resources. Because there are often more peers than there are different resource categories or n, a search along the transit ring network in [20] can be extremely effective. The difficulty involved in data lookup is a function of the number of peers N in the system in Chord [10] and other structured P2P systems [9,11,12], but it is a function of the number of different resource categories n in the proposed design [20]. The important thing to note is that the search procedure has been made easy and effective using the same code to indicate a resource type $R_i$ and the related group-head $C_i$. As a result, the time complexity for data search in the architecture shown in [20] is constrained by $(1 + \frac{n}{2})$. Table 1 shows the data search difficulty of both [20] strategy and other significant already-existing DHT-based systems.

**Table 1.** Data Lookup: Complexity Comparison.

| | Can | Chord | Pastry | RC-Based |
|---|---|---|---|---|
| Architecture | DHT-based | DHT-based | DHT-based | RC-Based |
| Lookup Protocol | {Key, value} pairs to map a point P in the coordinate space using uniform hash functions | Matching Key and NodeID | Matching Key and prefix in NodeID | Inter-Group: Routing through group heads Intra-Group: Complete Graph |
| Parameters | N number of peers in the network, d-number of dimensions | N number of peers in the network | N number of peers in the network, b-number of bits (B = 2b) use for the base of the chosen identifier | n = Number of distinct resource types, N-number of peers in the network, n $<<$ N |
| Lookup Performance | $O(dN^{1/d})$ | $O(logN)$ | $O_B(logN)$ | Inter-Group: $O(n)$, Intra-Group: $O(1)$ |

## 3. Publish/Subscribe and RC-Based P2P Architecture for Fog Computing

In this section, we suggest overlaying the actual fog nodes with two networking infrastructures to provide effective resource discovery. For our Healthcare 4.0 fog-computing

architecture, we have made the following modifications described below to employ the RC-based p2p architecture mentioned in Section 2.2.

First, the network engineer organizes the heterogeneous network in the form of groups according to meaningful *range of computing power*, for example, lower group logical address can cater to nodes with 1–2 cores and <256 MB RAM, whereas higher group logical address can cater to nodes with higher range can be 16–32 cores with a few GB RAM. This essentially refers to the capacity to complete the necessary operations and store the results. Assigning the logical address and the number of groups is at the *discretion of the designer based on the application needs*. This implies that the logical address value is not representative of the abstract value for the computation capacity i.e., for example, group $C_0$ has no compute cores never happens. This current scheme is a deviation from the prior work on interest-based group assignments. The range allows the heterogeneous network to accommodate minor changes in the group's needs. However, if it changes drastically, then the node will inform this information to its group-head, who will refer to the GRT and place the node in the group with the respective computing range.

Second, to incorporate this concept of peer computing power changing drastically and making data lookup efficient, we have included another column in the GRT which represent the computing range for each of the groups defined in our RC-based fog-computing architecture. The range i in our proposed architecture represents the resource $R_i$ in the existing RC-based architecture. An example, let us consider an RC-based p2p fog-computing network with 3 groups, and their respective computing ranges are defined by $R_0$, $R_1$, $R_2$, the respective GRT table is shown in Table 2. Now if the requested range falls in the boundary condition, for example between $R_1$ and $R_2$, the request will be forwarded to the group-head with a higher computing range $R_2$.

**Table 2.** Global Resource Table (GRT).

| Global Resource Table (GRT) | | |
|---|---|---|
| **Group-Head Logical Address** | **IP-Address** | **Computing Range** |
| 0 | 172.16.254.1 | $R_0$ |
| 1 | 160.15.244.4 | $R_1$ |
| 2 | 171.10.230.7 | $R_2$ |

Figure 2 depicts the fundamental architecture of a fog computing-based healthcare platform. This demonstrates the existence of a fog controller at the fog-computing layer. The fog controller can be utilized as the component in charge of gathering data from the underlying sensor layer. The fog controller will be informed about a group-head $C_x^h$ logical address and IP address belonging to the RC-based architecture. Once given the task, it forwards the request to a group-head $C_x^h$. $C_x^h$ searches for an appropriate fog node or group of fog nodes to do the required data analysis. Eventually, the output will be transmitted directly to the sensor layer actuators that are accessible for use or filtered data may be transferred to the cloud layer for additional processing or storage. Publish/subscribe and P2P overlays are the two communication paradigms used in our suggested network architecture.

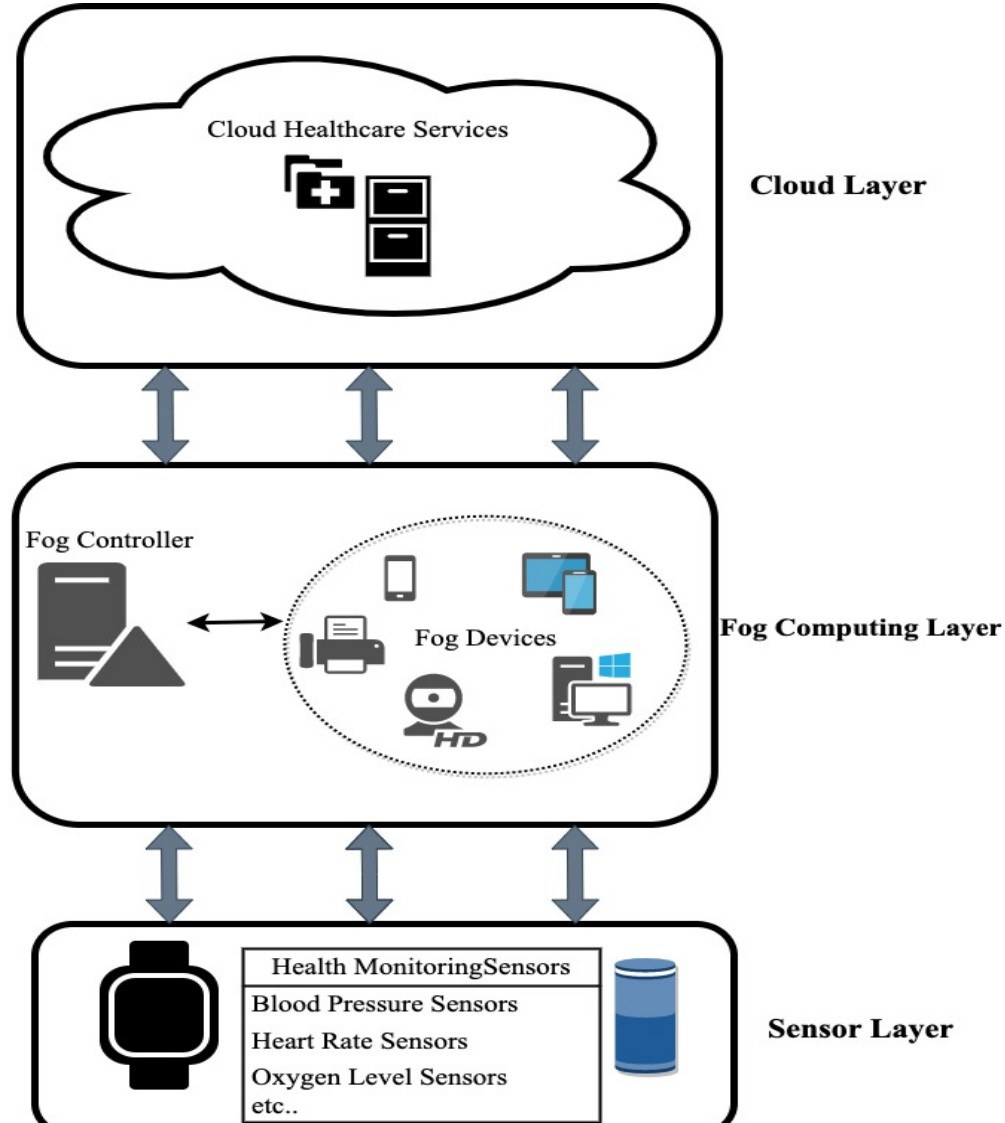

**Figure 2.** Healthcare 4.0: Fog-Computing Architecture.

### 3.1. PubSub-Based Fog-Computing Framework

For effective resource identification and work allocation, we first suggest using a publish/subscribe-based (PubSub-Based) communication model. In Figure 3, the proposed architecture is shown. The utilization of a fog controller serving as an event-notification agent for fog node p2p architecture is one of the main topics of this article. The fog controller may be duplicated in this framework to create a fault-tolerant system. The fog nodes in this architecture are arranged in groups as described earlier.

As mentioned in earlier, the group-heads in this architecture maintain the GRT (global resource table) containing information about the other group-heads present in the RC-based architecture, as well as LRT (local resource table) containing information about the other members in its group. The fog controller will be informed about a group-head $C_x^h$ logical address and IP address belonging to the RC-based architecture. To make things simple, it can be the information about the first group-head who has joined the architecture $C_0^h$, but again it can be any group-head belonging to the RC-based architecture. Additionally, the sensors that are serving as publishers provide the controller. Based on the sensor releasing the data, the fog controller is configured to determine the computing power needed to process the data. When a publication is available with the controller, it sends

the advertised job $ReR_i$ to the group-head $C_x^h$. The group-head $C_x^h$ will now follow the data-lookup protocol defined in Section 3.2.1.

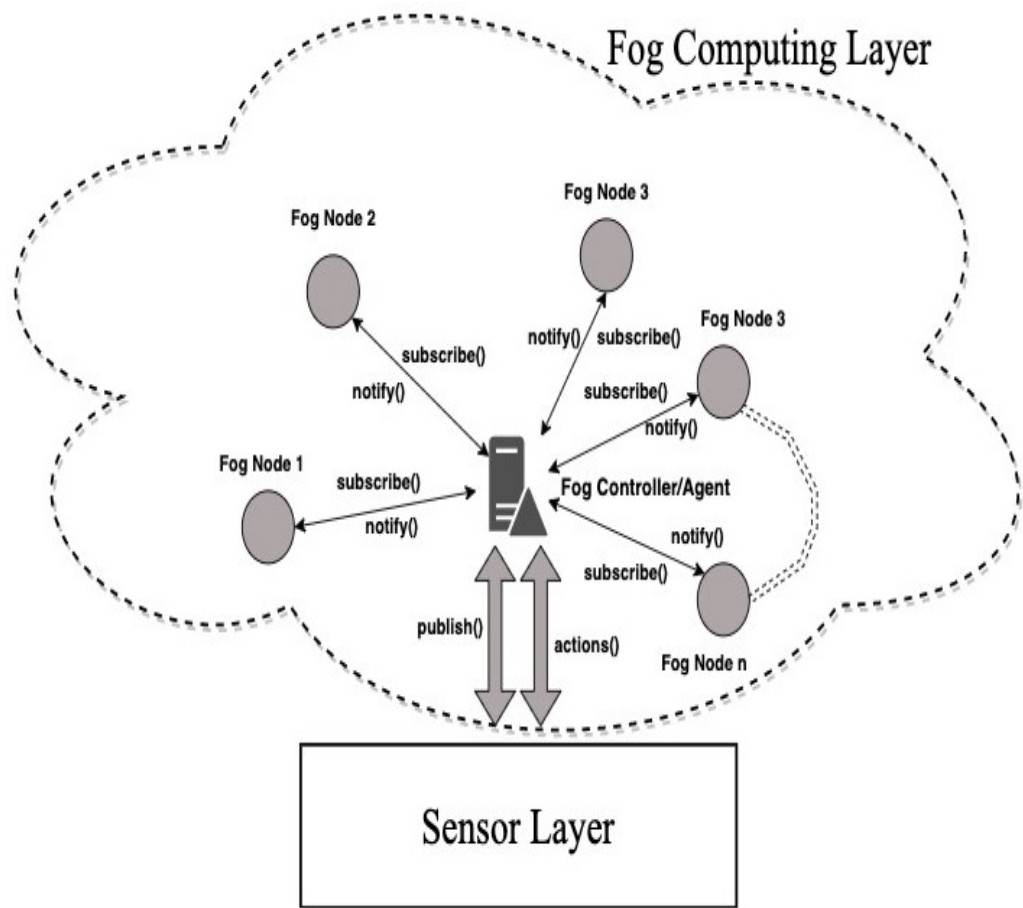

**Figure 3.** Publish/Subscribe Resource Discovery.

### 3.2. RC-Based P2P Fog-Computing Architecture

The three-layer architecture of RC-based P2P fog computing consists of client nodes/IoT devices in the first layer, fog nodes in the second layer, and a cloud node in the third layer. Between the client nodes/IoT devices and the fog nodes lies the fog controller. To link fog nodes and enable resource lookup and data transfers between them, RC-based P2P fog-computing architecture uses a residue class-based network model as discussed in Section 2.2. Fog nodes now employ RC-based lookup techniques to find resources and are connected over P2P. The architecture of RC-based peer-to-peer fog computing is shown in Figure 4. When a client node requests a resource or piece of data to the fog controller, based on the data, the fog controller is configured to determine the computing capacity needed $ReR_i$, it forwards the requests to its known group-head $C_x^h$, the services are maintained by implementing RC-based lookup protocols. The group-head $C_x^h$ will now perform the data-lookup protocol in the RC-based architecture. In that case, there are four different possibilities.

- **Scenario 1:** The fog node $C_x^h$ itself has $ReR_i$.
- **Scenario 2:** The fog node who has $ReR_i$ is the group member of the group $C_x$.
- **Scenario 3:** The fog node who has $ReR_i$ is the group-head of another group $C_y^h$.
- **Scenario 4:** The fog node who has $ReR_i$ is the group member of another group $C_y$.

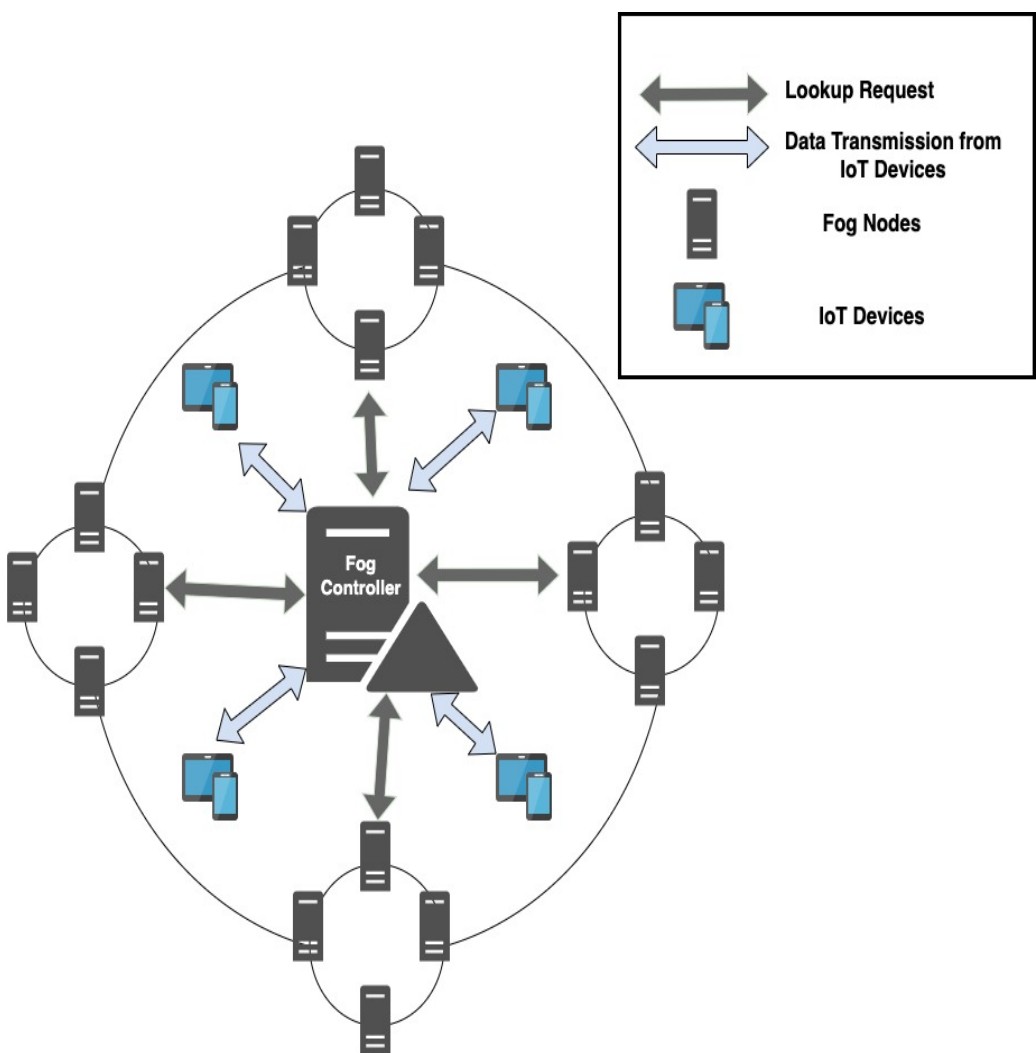

**Figure 4.** Pub/Sub and RC-Based P2P Fog-Computing Architecture.

### 3.2.1. RC-Based Lookup Protocol

Assume an RC-based peer-to-peer system with n different computing ranges, let S be the set of all peers. Then $S = \{C_i\}, (0 \leq i \leq n-1)$, where $C_i$ represents the subset of peers that have the computing power within the range $R_i$. This subset $C_i$ is referred to as group i in this study. Furthermore, we presume that $C_i^h$ is the first peer among the peers in each group $C_i$ to join the system. The group-head of group $C_i$ is referred to as $C_i^h$.

Assume the IoT/client device ($I_x$) sends some data to the fog controller (FC), based on the data, the FC is configured to determine the computing capacity needed $ReR_i$. For simplicity, we have considered that the fog controller FC is informed about the fog node $C_0^h$, which is the group-head of the computing range $R_0$ and also the first one to have joined the RC-based architecture. FC, after determining the computing capacity required, has forwarded the advertised job $ReR_i$ to $C_0^h$. $C_0^h$ will now perform the lookup service in the RC-based architecture. Here we present the algorithms for all the scenarios described in Section 3.2.

- **Scenario 1 (Section 3.2):** The fog node $C_0^h$ itself has $ReR_i$.

    – FC forwards the advertised job $ReR_i$ to $C_0^h$.
    – If $ReR_i$ falls in the computing range $R_0$ and the fog node $C_0^h$ have the resource $ReR_i$ it will provide service to FC.

This scenario is presented in Algorithm 1.

---

**Algorithm 1** Scenario 1 (Section 3.2): The fog node $C_0^h$ itself has the resource

---

1:  FC forwards the advertised job $ReR_i$ to $C_0^h$;
2:  **if** $(ReR_i \in R_0) \wedge C_0^h$ possess $ReR_i$ **then**
3:      $C_0^h$ unicasts service to FC;
4:  **else if** Scenario 2 or Scenario 3 or Scenario 4 **then**
5:      Respective Scenario solution
6:  **else**
7:      $C_0^h$ informs FC no one has $ReR_i$;
8:      cloud node is contacted by FC for service $ReR_i$;
9:      cloud responds with $ReR_i$ to FC;
10: **end if**

---

**Observation 1:** The number of hops required for an FC to find a resource in the proposed overlay P2P architecture for **Scenario 1 (Section 3.2)** is only 2 which is constant.

- **Scenario 2 (Section 3.2):** The fog node who has $ReR_i$ is the group member of the group of $C_0$.

  – FC forwards the advertised job $ReR_i$ to $C_0^h$.
  – Fog node $C_0^h$ does not have $ReR_i$,
  – If $ReR_i$ falls in the computing range $R_0$,it will broadcast the advertised message $ReR_i$ in its group $C_0$ using LRT.
  – Fog node $F_i$ in $C_0$ who has $ReR_i$ will reply with the service to the $C_0^h$.
  – $C_0^h$ will reply to FC with the service.

  This scenario is presented in Algorithm 2.

---

**Algorithm 2** Scenario 2 (Section 3.2): The fog node that has the resource is the group member of the group of $C_x$

---

1:  FC forwards the advertised job $ReR_i$ to $C_0^h$;
2:  **if** $(ReR_i \in R_0) \wedge C_0^h$ does not possess $ReR_i$ **then**
3:      $C_0^h$ broadcast the advertised message $ReR_i$ in its group $C_0$ using LRT;
4:      **if** $F_i \in C_0$ possess $ReR_i$ **then**
5:          $F_i$ unicasts service to $C_0^h$;
6:          $C_0^h$ respond to FC with the service $ReR_i$;
7:      **else if** Scenario 3 or Scenario 4 **then**
8:          Respective Scenario solution
9:      **else**
10:         $C_0^h$ informs FC no one has $ReR_i$;
11:         cloud node is contacted by FC for service $ReR_i$;
12:         cloud responds with $ReR_i$ to FC;
13:     **end if**
14: **end if**

---

**Observation 2:** The number of hops required for an FC to find a resource in the proposed overlay P2P architecture for **Scenario 2 (Section 3.2)** is only 4 which is constant.

- **Scenario 3 (Section 3.2):** The fog node that has the resource is the group-head of another resource type $C_y^h$.

  – FC forwards the advertised job $ReR_i$ to $C_0^h$.
  – Fog node $C_0^h$ and any $F_i \in C_0$ does not have the resource $ReR_i$.
  – $C_0^h$ determines the group-head $C_y^h$'s address code from GRT such that $ReR_i \in$ the computing range $R_y$ for $C_y^h$ (i = y).
  – $C_0^h$ computes $|i - j| = h$.

- Based upon the value of h, it will forward $ReR_i$ to its predecessor or its successor.
- Every group-head traversed $C_i^h$ forwards $ReR_i$ until i = y.
- If $C_i^h$ has the resource, it will reply with the service to the $C_0^h$.
- $C_0^h$ will reply to the FC with the service.

This scenario is presented in Algorithm 3.

---

**Algorithm 3** Scenario 3 (Section 3.2): The fog node that has the resource is the group-head of another resource type $C_y^h$

---

1: FC forwards the advertised job $ReR_i$ to $C_0^h$;
2: **if** $ReR_i \notin R_0$ **then**
3:    $C_0^h$ determines the group-head $C_y^h$'s address code from GRT;        ▷ $ReR_i \in$ the computing range $R_y$ for $C_y^h$ ( i = y)
4:    $C_0^h$ computes $|i - j| = h$;
5:    **if** h > n/2 **then**          ▷ n = total no. of computing ranges
6:       $C_0^h$ forwards the advertised message $ReR_i$ along with $C_y^h$'s IP address to its predecessor $C_{n-1}^h$;
7:    **else**
8:       $C_0^h$ forwards the advertised message $ReR_i$ along with $C_y^h$'s IP address to its successor $C_1^h$;    ▷ Looking for minimum no. of hops along the transit ring network
9:    **end if**
10:   All intermediate group-heads $C_i^h$ forwards until i = y ▷ no. of hops along the ring in the worst case is n / 2
11:   **if** $C_y^h$ possess $ReR_i$ **then**
12:      $C_y^h$ unicasts service to $C_0^h$;
13:      $C_0^h$ respond to FC with the service $ReR_i$;
14:   **else if** Scenario 4 **then**
15:      Respective Scenario solution
16:   **else**
17:      $C_0^h$ informs FC no one has $ReR_i$;
18:      cloud node is contacted by FC for service $ReR_i$;
19:      cloud responds with $ReR_i$ to FC;
20:   **end if**
21: **end if**

---

**Observation 3:** The number of hops required for an FC to find a resource in the proposed overlay P2P architecture for **Scenario 3 (Section 3.2)** is $n + 2$, where n is the total number of computing ranges in the network, the data-lookup complexity is $O(n)$.

- **Scenario 4 (Section 3.2):** The fog node who has $ReR_i$ is the group member another group $C_y$.

  - FC forwards the advertised job $ReR_i$ to $C_0^h$.
  - Fog node $C_0^h$ and any $F_i \in C_0$ does not have $ReR_i$.
  - $C_0^h$ determines the group-head $C_y^h$'s address code from GRT such that $ReR_i \in$ the computing range $R_y$ for $C_y^h$ (i = y).
  - $C_0^h$ computes $|i - j| = h$.
  - Based upon the value of h, it will forward $ReR_i$ to its predecessor or its successor.
  - Every group-head traversed $C_i^h$ forwards $ReR_i$ until i = y.
  - If $C_y^h$ does not have $ReR_i$ it will broadcast the message $ReR_i$ in group $C_y$.
  - Fog node $F_y$ in $C_y$ who has $ReR_i$ will reply with the service to the $C_y^h$.
  - $C_y^h$ will reply to $C_0^h$ with the service.
  - $C_0^h$ will reply to FC with the service.

This scenario is presented in Algorithm 4.

---

**Algorithm 4** Scenario 4 (Section 3.2): The fog node that has $ReR_i$ is the group member of the group of another resource type $C_y$

---

1: FC forwards the advertised job $ReR_i$ to $C_0^h$;
2: **if** $ReR_i \notin R_0$ **then**
3:   $C_0^h$ determines the group-head $C_y^h$'s address code from GRT;            ▷ $ReR_i \in$ the computing range $R_y$ for $C_y^h$ ( i = y)
4:   $C_0^h$ computes $|i - j| = h$;
5:   **if** h > n/2 **then**                               ▷ n = total no. of computing ranges
6:     $C_0^h$ forwards the advertised message $ReR_i$ along with $C_y^h$'s IP address to its predecessor $C_{n-1}^h$;
7:   **else**
8:     $C_0^h$ forwards the advertised message $ReR_i$ along with $C_y^h$'s IP address to its successor $C_1^h$;     ▷ Looking for minimum no. of hops along the transit ring network
9:   **end if**
10:   All intermediate group-heads $C_i^h$ forwards until i = y ▷ no. of hops along the ring in the worst case is n / 2
11:   **if** $C_y^h$ does not possess $ReR_i$ **then**
12:     $C_y^h$ broadcast the advertised message $ReR_i$ in its group $C_y$ using LRT;
13:     **if** $F_y \in C_y$ possess $ReR_i$ **then**
14:       $F_y$ unicasts service to $C_y^h$;
15:       $C_y^h$ unicasts service to $C_0^h$
16:       $C_0^h$ respond to FC with the service $ReR_i$;
17:     **else**
18:       $C_0^h$ informs FC no one has $ReR_i$;
19:       cloud node is contacted by FC for service $ReR_i$;
20:       cloud responds with $ReR_i$ to FC;
21:     **end if**
22:   **end if**
23: **end if**

---

**Observation 4:** The number of hops required for an FC to find $ReR_i$ in the proposed overlay P2P architecture for **Scenario 4 (Section 3.2)** is $n + 4$, where n is the total number of resources in the network, the data-lookup complexity is $O(n)$.

## 4. Performance Evaluation

In this section, we discuss the performance of the proposed RC-based Lookup Protocol Algorithm proposed in Section 3.2.1.

In [5] a DHT-based fog-computing architecture has been proposed for Healthcare 4.0. Specifically, the authors have taken the example of Chord [10] architecture to solve the problem. As the average number of hops needed for each search in Chord is N/2, where N is the total number of peers in the system, following the chord is not suitable since it is exceedingly inefficient in big peer-to-peer systems.

The complexity of the data-lookup protocol in the current study is constant in the best-case scenario, i.e., Scenario 1 and 2, and is $O(n)$ as discussed in Section 3.2.1 for the other scenarios. In contrast to the studies in [5], search latency is independent of the total number of peers in the network. Because of its much lower search latency, we deduce that the suggested structured interest-based overlay architecture might be preferred over non-interest-based structured ones. Here, by interest, we refer to the resources of different computing ranges as described earlier.

The important thing to note is that the search procedure has been made easy and effective using the computing ranges to indicate a resource type $R_i$ and the related group-head $C_i^h$. In addition, the size of the GRT database is only constrained by the variety of

computing ranges, which is very small. In actuality, there are many more peers than there are different computing ranges, and the size of the GRT is independent of the total number of peers N in the P2P system. The freshly joining group-head will always have the greatest logical address since GRT expands dynamically as additional group-heads join. Because GRT is always ordered in relation to the logical addresses of the group-heads, searching inside it is particularly effective.

*Experiments*

As we already indicated, the major goal of the current study is to demonstrate how our interest-based, non-DHT design outperforms DHT-based systems in terms of search latency and data-lookup complexity. We thus conducted three tests to evaluate the data-lookup latency on all the scenarios mentioned in Section 3.2.1 in terms of overlay hops of the RC-based Fog computing p2p architecture with a well-known p2p architecture Chord [10], in addition to the analytical comparison. Results of the experiments with three different numbers of distinct resource types are shown in Figures 5–7.

- **Scenario 1:**

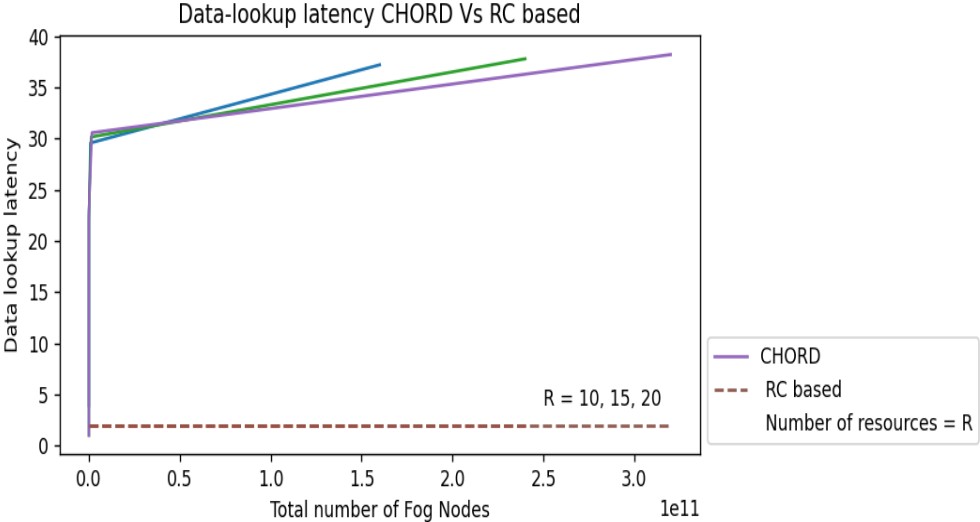

**Figure 5.** Scenario 1: Chord vs. RC-based.

- **Scenario 2:**

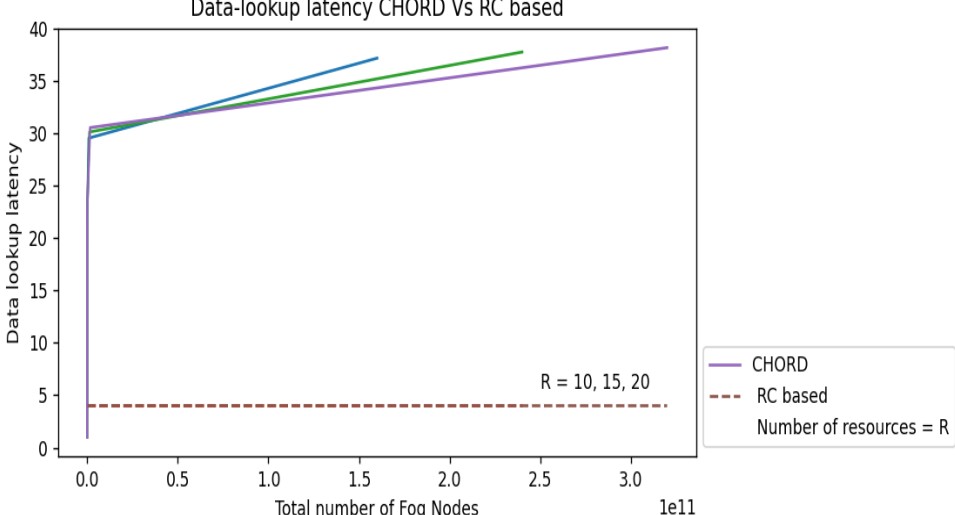

**Figure 6.** Scenario 2: Chord vs. RC-based.

- **Scenario 3 and 4:**

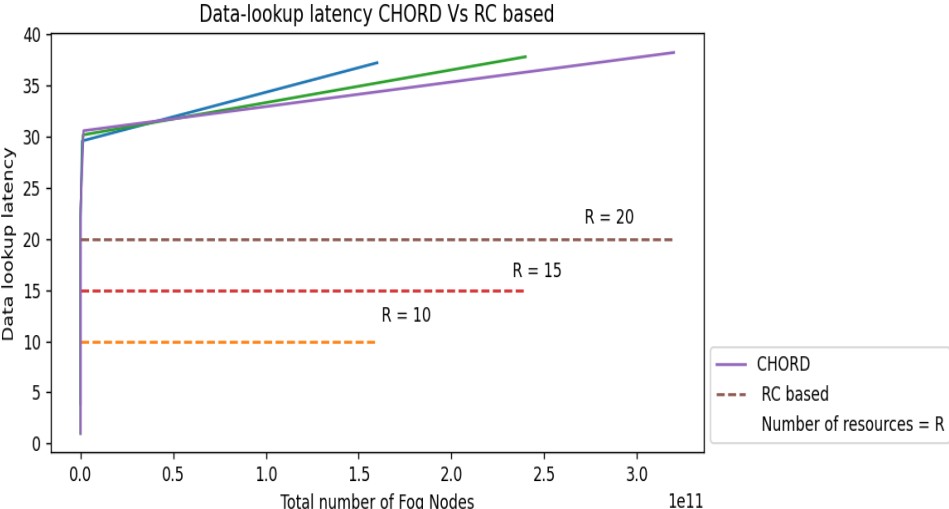

**Figure 7.** Scenario 3 and 4: Chord vs. RC-based.

We have taken into account RC-based overlay networks with 10, 15, and 20 different resource kinds; the number of fog nodes in each of the groups corresponding to the different computing ranges has steadily risen, culminating in a final system fog node count of $3 \times 10^{11}$ for all the scenarios. Consider N as the total number of fog nodes in the overlay network. For Chord [10] the data-lookup latency is $O(log_2 N)$. The data search latency for Chord grows with the number of fog nodes in the system, as seen in each picture since this delay is dependent on the total number of fog nodes N in the system. The data search latency in the RC-based design, however, is unaffected by the size of any individual group or the system's total number of fog nodes N. As a result, the RC-based design will have a constant data-lookup latency for Scenarios 1 and 2 and n for Scenarios 3 and 4, where n is the total number of resources n $<<$ N, as seen in each of the two images as a straight line with zero gradients as the number of fog nodes in the system grows. In each of the two pictures, we can see a considerable reduction in data-lookup latency compared to Chord [10] due to our suggested design.

## 5. Conclusions

Healthcare 4.0 heralds a transition in the healthcare sector from manual processes to decisions based on technology. This change intends to make greater use of patient experience technology. A variety of IoT-based devices are employed to continually collect patient health data. Initially, these data were processed in cloud-based data centers. Issues with latency, privacy, and cost were present in this method. Fog computing has provided a solution to the problems associated with cloud-based data processing. Fog computing also has its own difficulties in terms of finding and making use of available resources.

In this study, we introduce resource/computing range-based, RC-based fog computing, a novel method for constructing a scalable Healthcare 4.0 fog-computing system that performs data-lookup operations with a high degree of efficiency, therefore lowering latency and delivering more rapid localized services. Because RC-based fog can handle greater processing needs locally, there are much fewer requests that must be routed to the cloud, which effectively saves Internet bandwidth. In the RC-based fog-computing concept, several cloud service types, including SaaS, PaaS, and IaaS, may be grouped at various level 2 networks. In our upcoming research, we will take into account the geographic location parameter to include numerous fog controllers supporting many IoT devices connecting to various group-heads in the RC-based fog-computing architecture to offer service. The suggested RC-based fog architecture will be further expanded to support multi-layer hierarchical Healthcare 4.0 P2P fog structures, and its performance

will be compared to that of current fog architecture. Additionally, in our immediate future research we plan to simulate our proposed architecture using the fog simulators such as IFogSim2 [24] and have real-time data generated to provide the architecture's effectiveness in realistic Healthcare 4.0 scenarios.

**Author Contributions:** Conceptualization, I.R. and R.M.; methodology, I.R. and B.G.; software, R.M.; validation, R.M. and N.R.; formal analysis, R.M.; investigation, I.R.; writing—review and editing, I.R.; visualization, R.M. and I.R.; supervision, B.G.; project administration, B.G. and N.R. All authors have read and agreed to the published version of the manuscript.

**Funding:** This research received no external funding.

**Data Availability Statement:** Not applicable.

**Acknowledgments:** We feel grateful to the reviewers for their insightful comments and recomendation which have allowed us to increase the manuscripts quality.

**Conflicts of Interest:** The authors declare no conflict of interest.

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
