# Peer review of "Efficient Non-DHT-Based RC-Based Architecture for Fog Computing in Healthcare 4.0"

_2624-831X, doi:10.3390/iot4020008_

Round 1

Reviewer 1 Report

The paper proposes a fog computing-based platform for healthcare that employs publish/subscribe and peer-to-peer (P2P) overlays to facilitate resource discovery. The architecture includes a fog controller at the fog computing layer that collects data from the underlying sensor layer and searches for a suitable fog node or group of fog nodes to perform the necessary data analysis. The authors suggest using an RC-based P2P fog computing architecture that employs residue class-based network models and RC-based lookup techniques to find resources and connect fog nodes. The proposed RC-based lookup protocol algorithm outperforms DHT-based systems regarding search latency and data lookup complexity. The RC-based fog computing architecture is scalable and efficient, reducing latency and faster-localized services. 

The proposed architecture can handle more significant processing requirements locally, effectively conserving internet bandwidth. In future research, the authors plan to expand the presented RC-based fog architecture to support multi-layer hierarchical Healthcare 4.0 P2P fog structures and compare its performance to current fog architecture.

The related works section is well-written and covers relevant topics on using smart devices in healthcare and the associated challenges. The section also explains the use of fog computing and publish/subscribe systems as practical solutions to overcome some of these challenges.

The paper is well-structured and analyzes fog computing architectures and communication protocols for creating a fog computing-based healthcare platform.

However, the paper could benefit from improved form, as there are English grammar and usage errors. Additionally, DHT and RC are used before being properly defined or introduced. Furthermore, line 229 has an incorrect reference: [?].

Suggested improvements for the authors: The authors could provide greater technical detail on the proposed architecture, including cluster configuration and node communication modes. Further exploration of the architecture's limitations and challenges, such as scalability and fault tolerance, would enhance the paper's contribution. The paper could also consider the impact of geographic factors on node distribution and resource availability. Finally, testing the proposed architecture's effectiveness in realistic Healthcare 4.0 scenarios would provide additional insights into its potential applications.

Author Response

Response to Reviewer 1 Comments

Thank you so much for your constructive comments. We have thus scrutinized the manuscript and corrected the grammar errors and typos. The manuscript has undergone a thorough revision according to the editor and reviewers’ comments. We have included all the comments of all 4 reviewers, as pointed out in the manuscript, the highlighted portions along with the track changes. However, we have not incorporated very few points because we will include those in our future research. We have stated these in the conclusions section of the article. Please see below our responses. 

Reviewer Comment 1.1 However, the paper could benefit from improved form, as there are English grammar and usage errors. Additionally, DHT and RC are used before being properly defined or introduced.
Reply:
Basic introduction about DHT and RC has been added in Section 1, line – 69 - 79.

Reviewer Comment 1.2 The authors could provide greater technical detail on the proposed
architecture, including cluster configuration and node communication modes.
Reply:
1. We have added a new section 2.2.1 - Relevant Properties of Modular Arithmetic, line - 215
2. For technical detail on the proposed architecture, including cluster configuration and node communication modes we have added contents in section 2.2.2 - Assignments of Overlay Addresses, line 239 - 273.

Reviewer Comment 1.3 — Further exploration of the architecture’s limitations and challenges, such as scalability and fault tolerance, would enhance the paper’s contribution.
Reply:
1. We have added a new section 2.2.4 - Fault Tolerance of the Architecture, line - 293.
2. We have added a new section 2.2.5 - Scalability of the Architecture, line - 311.

Reviewer Comment 1.4 — The paper could also consider the impact of geographic factors on node distribution and resource availability.
Reply:
1. We addressed this comment in section 5 - Conclusions, line - 543.

Reviewer Comment 1.5 — Finally, testing the proposed architecture’s effectiveness in realistic Healthcare 4.0 scenarios would provide additional insights into its potential applications.
Reply:
1. We addressed this comment in section 5 - Conclusions, line - 548.

Reviewer 2 Report

This paper proposes a novel 2-level overlay network to address the efficiency challenges in healthcare fog computing systems. The paper is well-structured and well-written.  Here are some comments.

 1. It is recommended to provide definitions when introducing new terms for the first time. e.g., RC, DHT. 

2. It is not clear for the definition of the resource in line 177.  Do you only consider the "computing power", or are there any other resource types? 

3. Regarding the resource type of "computing power",  how do you categorize this term to several values?  One more question is, what if the current computing power change from one value to another value, will the algorithm capture the changes, and re-calculate again? 

3. In line 197, as the authors mentioned,  Resource Code is the same as the group-head logical address. Why do we specify duplicate information in GRT tuple <Group Head Logical address, IP address, Resource Code>? 

4. Please add more details on the modular algorithm mentioned in line 226 & 227. 

5. Some typos need to be fixed. e.g, Line 229. 

Author Response

Response to Reviewer 2 Comments

Thank you so much for your constructive comments. We have thus scrutinized the manuscript and made the necessary corrections and modifications according to the comments. The manuscript has undergone a thorough revision according to the editor and reviewers’ comments. We have included all the comments of all 4 reviewers, as pointed out in the manuscript, the highlighted portions along with the track changes. However, we have not incorporated very few points because we will include those in our future research. We have stated these in the conclusions section of the article. Please see below our responses. 

Reviewer Comment 2.1 It is recommended to provide definitions when introducing new terms for the first time. e.g., RC, DHT.
Reply: Basic introduction about DHT and RC has been added in Section 1, line – 69.

Reviewer Comment 2.2 It is not clear for the definition of the resource in line 177. Do you only consider the ”computing power”, or are there any other resource types?
Reply: We addressed this comment in section 3 - Publish/Subscribe and RC Based P2P Architecture for Fog Computing , line - 344.

Reviewer Comment 2.3 Regarding the resource type of ”computing power”, how do you categorize this term to several values? One more question is, what if the current computing power change from one value to another value, will the algorithm capture the changes, and re-calculate again?
Reply: We addressed this comment in section 3 - Publish/Subscribe and RC Based P2P Architecture for Fog Computing , line - 344.

Reviewer Comment 2.4 — In line 197, as the authors mentioned, Resource Code is the same as the group-head logical address. Why do we specify duplicate information in GRT tuple <Group Head Logical address, IP address, Resource Code>?
Reply: We addressed this comment in section 2.2 - RC-Based P2P Architecture, line - 204 - 208.

Reviewer Comment 2.5 — Please add more details on the modular algorithm mentioned in line 226 & 227.
Reply: We addressed this comment in section 2.2.1. Relevant Properties of Modular Arithmetic, line - 215.

Reviewer Comment 2.6 — Some typos need to be fixed. e.g, Line 229.
Reply: We addressed this comment and did the necessary typo corrections.

Reviewer 3 Report

This paper presents a publish/subscribe and interest/resource based non-DHT based peer-to-peer (P2P) RC-based architecture for resource discovery for healthcare IOT applications. A 2-level overlay network is proposed which consists of (1) transit ring containing group heads representing a particular resource type, and (2) completely connected group of peers. This work aims to shorten the resource search latency.

Basically, the proposed RC-based P2P is based on their previous work in ref [19].

However, whether it is appropriate to apply a P2P overlay network to solve the healthcare IOT problem is still debatable. The purpose of P2P is normally targeting at the reducing the high volume demand of the cloud servers (like streaming videos) by facilitating peers sharing. The user churn caused by user join or leave is the major reason why keeps the data booking difficult. But in the healthcare environment, the data amount generated by sensors could be relatively small. The server-throughput bottleneck does not seem to be a major issue. Most fog nodes (peers) are dedicated devices and quite stable. User churn is not a serious issue. The resource search issue that this paper tries to address, I believe, should be sufficiently handled by few central servers. In summary, more rigorous arguments should be made to convince the audience the necessity of applying P2P networks.

The technical writing of this paper should be significantly improved. Definitions or notations are quite confused which make the paper hard to read. The paper should be thoroughly proofread before submission. Some examples are given as follows.

1.      In section 2.2, the description of RS P2P is very confused. It should be totally rewritten. For instance, Definition 1 the notation of the tuple is confused (line 175). The definition of S=C_i is confused (line 181). The notations of clusters or group are ambiguous in line 184 and 190.

2.      Line 214 give Definition 1 once again? Also this definition only gives one direction of link. For example h8 connects to h9. How about the other way around? (i.e., h9 connects to h8).

3.      Lemmas stated in Line 220 and 221 seem very obvious. Usually a lemma states some results that are not so obvious and require simple proof.

4.      Line 252 says “ As a result, the time complexity for data search in the architecture we have shown is constrained by 1 n.” Why? Should be explained.

5.      In Line 267, “3.1. PubSub-Based”. Is it a typo?

6.      Again, the definition of S=C_i is confused (line 316).

7.      Dangling references can be found in many places (e.g., lines 69, 117, 229).

Author Response

Response to Reviewer 3 Comments

Thank you so much for your constructive comments. We have thus scrutinized the manuscript and made the necessary corrections and modifications according to the comments. The manuscript has undergone a thorough revision according to the editor and reviewers’ comments. We have included all the comments of all 4 reviewers, as pointed out in the manuscript, the highlighted portions along with the track changes. However, we have not incorporated very few points because we will include those in our future research. We have stated these in the conclusions section of the article. Please see below our responses.   

Reviewer Comment 3.1 — However, whether it is appropriate to apply a P2P overlay network to solve the healthcare IOT problem is still debatable. The purpose of P2P is normally targeting at the reducing the high volume demand of the cloud servers (like streaming videos) by facilitating peers sharing. The user churn caused by user join or leave is the major reason why keeps the data booking difficult. But in the healthcare environment, the data amount generated by sensors could be relatively small. The server-throughput bottleneck does not seem to be a major issue. Most fog nodes (peers) are dedicated devices and quite stable. User churn is not a serious issue. The resource search issue that this paper tries to address, I believe, should be sufficiently handled by few central servers. In summary, more rigorous arguments should be made to convince the audience the necessity of applying P2P networks.
Reply: We addressed this comment in section 1 - Introduction, line - 95 - 115.

Reviewer Comment 3.2 — In section 2.2, the description of RS P2P is very confused. It should be totally rewritten. For instance, Definition 1 the notation of the tuple is confused (line 175). The definition of = C
i is confused (line 181). The notations of clusters or group are ambiguous in line 184 and 190.
Reply: We addressed this comment in section 2.2. RC-Based P2P Architecture, line - 185 - 196.

Reviewer Comment 3.3 Line 214 give Definition 1 once again? Also this definition only gives one direction of link. For example h8 connects to h9. How about the other way around? (i.e., h9 connects to h8).
Reply: We addressed this comment in section 2.2. RC-Based P2P Architecture, line - 261. The links are bi-directional.

Reviewer Comment 3.4 Lemmas stated in Line 220 and 221 seem very obvious. Usually a lemma states some results that are not so obvious and require simple proof.
Reply: We addressed this comment in section 2.2 - RC-Based P2P Architecture, line - 267 - 273.

Reviewer Comment 3.5 Line 252 says “ As a result, the time complexity for data search in the architecture we have shown is constrained by 1 n.” Why? Should be explained.
Reply: We addressed this comment in section 2.3. Comparison of Distributed Hash Table P2P vs RC-Based P2P, line - 329 - 342.

Reviewer Comment 3.6 In Line 267, “3.1. PubSub-Based”. Is it a typo?
Reply: It is a short name that we have given for the publish and subscribe based system. We have addressed this in section 3.1 PubSub-Based Fog-Computing Framework, line 343.

Reviewer Comment 3.7 Again, the definition of S = Ci is confused (line 316).
Reply: We have addressed this in section 3.2.1. RC-based lookup protocol, line 420.

Reviewer Comment 3.8 Dangling references can be found in many places (e.g., lines 69,117, 229).
Reply: We have made the necessary corrections for the dangling references in the document.

Reviewer 4 Report

Healthcare 4.0 heralds a transition in the healthcare sector from manual processes to decisions based on technology. Fog computing has provided a solution to the problems associated with cloud-based data processing.

 In this paper introduce resource-based, RC-based fog computing, a novel method for constructing a scalable Healthcare 4.0 fog computing system that performs data lookup operations with a high degree of efficiency, therefore lowering latency and delivering more rapid localized services.

 The experimental data need to be improved and strengthened.

Accept after modification.

Author Response

Response to Reviewer 4 Comments

Thank you so much for your constructive comments. We have thus scrutinized the manuscript and made the necessary corrections and modifications according to the comments. The manuscript has undergone a thorough revision according to the editor and reviewers’ comments, as pointed out in the manuscript, the highlighted portions along with the track changes. However, we have not incorporated very few points because we will include those in our future research. We have stated these in the conclusions section of the article. Please see below our responses.   

Reviewer Comment 4.1 — The experimental data need to be improved and strengthened.
Reply: In this article we have generated a analytical experimental data based upon the data lookup latency for our proposed architecture and CHORD p2p architecture. The suggested RC-based fog architecture will be further expanded to support multi-layer hierarchical Healthcare 4.0 P2P fog structures,
and its performance will be compared to that of current fog architecture. Also, in our immediate future research we plan to simulate our proposed architecture using the fog simulators like IFogSim2 and have real-time data generated to provide the architecture’s effectiveness in realistic Healthcare 4.0 scenarios.
We have addressed this in section 5 - Conclusions, line - 545 - 550.

Round 2

Reviewer 3 Report

The comments were not fully addressed. The paper is still hard to read in this present form. Besides, I think the most fundamental question is the appropriateness of applying P2P to solve the issue raised by the paper. The authors should clear the doubt of the readers, instead of just claiming what they believe.

Author Response

Improving data locality in P2P-based fog computing is currently a very prominent research area, especially with its application in the area of Health Care [1]. We suggest reviewer 3 to please refer to [1] which gives an overview of the state-of-the-art research work that is going on now in the area of ‘Fog computing in health care’ . As mentioned in our last comment, we have addressed this comment in section 1 - Introduction, lines - 95 - 115. 

References:
1. Shukla, N., Gandhi, C. (2021). Efficient Resource Discovery and Sharing Framework for Fog Computing in Healthcare 4.0. In: 602 Tanwar, S. (eds) Fog Computing for Healthcare 4.0 Environments. Signals and Communication Technology. Springer, Cham. 603